# A Tool Bottleneck Framework for Clinically-Informed and Interpretable Medical Image Understanding

**Christina Liu**[*1,2]                                                   CLIU7@CALTECH.EDU
**Alan Q. Wang**[*2]                                                  ALANQW@STANFORD.EDU
**Joy Hsu**[2]                                                          JOYCJ@STANFORD.EDU
**Jiajun Wu**[2]                                                   JIAJUNWU@CS.STANFORD.EDU
**Ehsan Adeli**[2]                                                    EADELI@STANFORD.EDU
[1] *California Institute of Technology, Pasadena, CA 91125*
[2] *Stanford University, Stanford, CA 94305*

**Editors:** Accepted for publication at MIDL 2026

## Abstract

Recent tool-use frameworks powered by vision-language models (VLMs) improve image understanding by grounding model predictions with specialized tools. Broadly, these frameworks leverage VLMs and a pre-specified toolbox to decompose the prediction task into multiple tool calls (often deep learning models) which are composed to make a prediction. The dominant approach to composing tools is using text, via function calls embedded in VLM-generated code or natural language. However, these methods often perform poorly on medical image understanding, where salient information is encoded as spatially-localized features that are difficult to compose or fuse via text alone. To address this, we propose a tool-use framework for medical image understanding called the Tool Bottleneck Framework (TBF), which composes VLM-selected tools using a learned Tool Bottleneck Model (TBM). For a given image and task, TBF leverages an off-the-shelf medical VLM to select tools from a toolbox that each extract clinically-relevant features. Instead of text-based composition, these tools are composed by the TBM, which computes and fuses the tool outputs using a neural network before outputting the final prediction. We propose a simple and effective strategy for TBMs to make predictions with any arbitrary VLM tool selection. Overall, our framework not only improves tool-use in medical imaging contexts, but also yields more interpretable, clinically-grounded predictors. We evaluate TBF on tasks in histopathology and dermatology and find that these advantages enable our framework to perform on par with or better than deep learning-based classifiers, VLMs, and state-of-the-art tool-use frameworks, with particular gains in data-limited regimes. The project details and the code are available at https://christinaliu2020.github.io/tbm/.

**Keywords:** Tool use, domain knowledge, interpretability, data-efficiency.

## 1. Introduction

Human clinical decision-making from medical images typically involves using domain knowledge to analyze or extract multiple clinically-relevant, often spatially-localized features and subsequently integrating them to make a prediction (such as diagnosis). For example, in dermatology, pigment networks and border irregularity are important in diagnosing malignant melanoma (Anantha et al., 2004; Stolz, 1994). As another example, nuclei count and

---

\* Contributed equally

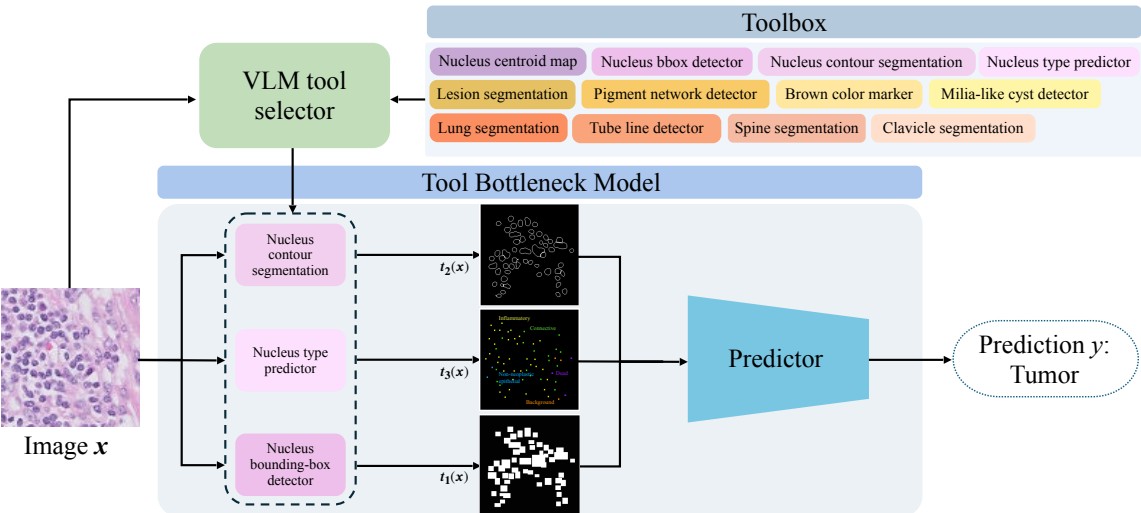

Figure 1: Overview of our proposed Tool Bottleneck Framework. A VLM selects tools from a pre-specified toolbox of clinically-relevant tools. These tools are passed to a Tool Bottleneck Model, which composes/fuses the tool outputs to make a prediction.

irregularity are known to be correlated with tissue pathology (Kuenen-Boumeester et al., 1984). For any given task, two questions arise: (1) which features are most relevant, and (2) how should these features be integrated to make a prediction?

Broadly, deep learning approaches leveraging general-purpose architectures such as convolutional neural networks (CNNs) and vision transformers (ViTs) address these two questions in a single unified pipeline, taking the raw image as input and training highly parameterized feature extractors and task predictors in an end-to-end fashion. These architectures provide the backbone for both image classification models and more recent vision-language models (VLMs) (Wang et al., 2022; Sellergren et al., 2025). However, they typically require large amounts of data to perform well due to their high number of learnable parameters. Additionally, they may be more challenging to interpret due to their end-to-end, black-box nature.

A promising line of work for overcoming the limitations of black-box deep learning includes *tool-use frameworks*. Broadly, tool-use frameworks leverage VLMs and a pre-specified toolbox to decompose the prediction task into multiple tool calls, which are then composed to make a prediction. Approaches like VisProg (Gupta and Kembhavi, 2023) and ViperGPT (Surís et al., 2023) perform visual understanding by using VLMs to output text (e.g., natural language or code) that composes tools. These tools are drawn from an extensive toolbox of modules that extract image-level and pixel-level features, such as segmentation or detection. However, these methods often fail to perform well on medical images, where text-based composition lacks the granularity for fusing fine-grained, spatially-localized features.

To address this, we propose the Tool Bottleneck Framework (TBF), a tool-use framework for medical image understanding that composes VLM-selected and clinically-informed tools with a learned Tool Bottleneck Model (TBM). Our framework is illustrated in Figure 1. First, for a given task and image $x$, a VLM selects the best tools from a pre-specified toolbox. This toolbox consists of multiple tools that each extract clinically-relevant features from a given image and modality. For example, the toolbox might include tools for histopathology, like nucleus segmentation or a cell typing tool, as well as tools for dermatology, like lesion segmentation or pigment network detection. Second, our proposed TBM takes as input $x$ and the VLM-selected tools and passes $x$ through each tool; the tools outputs are fused by a learned neural network which outputs a prediction. Inspired by Concept Bottleneck Models (Koh et al., 2020), TBMs flow information through a bottleneck layer, but generalize this layer to handle both spatially-localized features at the pixel-level as well as image-level scalar attributes.

We find that TBF not only overcomes limitations of state-of-the-art tool-use frameworks, but also yields predictors that are more interpretable and clinically-grounded. In particular, ours is the first work to incorporate clinically-grounded priors for tool use using a neural-network based composition. Our contributions are as follows:

- We propose the Tool Bottleneck Framework (TBF) for clinically-informed and interpretable tool-use for medical image understanding, which uses a medical VLM to select the tools most relevant for the image and task, then composes the selected tools to make a prediction with a learned Tool Bottleneck Model (TBM).
- We propose a simple yet effective strategy for TBMs to handle arbitrary tool selections.
- In tasks derived from histopathology and dermatology, we demonstrate on-par to superior performance compared to state-of-the-art baselines while being more interpretable, with particular gains in data-limited regimes.

## 2. Related Works

CNNs (Tan and Le, 2019; He et al., 2016; Huang et al., 2017; Szegedy et al., 2015; Howard et al., 2017) and ViTs (Dosovitskiy et al., 2021; Touvron et al., 2021; Liu et al., 2021) remain the dominant backbone for medical image diagnosis. These black-box, highly-parametrized architectures achieve strong performance but are known to be data-hungry (Sun et al., 2017; Bello et al., 2022) and lack interpretability (Selvaraju et al., 2017). More recently, medical vision-language models (VLMs) have further increased data scale, leveraging large-scale pretraining on image-text pairs to enable flexible zero- and few-shot generalization for report generation, diagnosis, and visual question answering (Ryu et al., 2025) across multiple tasks and modalities. Examples include MedCLIP (Wang et al., 2022), BioMedCLIP (Zhang et al., 2024), EyeCLIP (Shi et al., 2025), and MedGemma (Sellergren et al., 2025).

Our work is inspired by Concept Bottleneck Models (CBMs) (Koh et al., 2020), which predict a concept layer consisting of multiple image-level concepts for a given image; these concepts are then fused to arrive at a final prediction. Thus, information flow is effectively "bottlenecked" through the concept layer. CBMs have been widely explored in medical imaging contexts (Marcinkevics et al., 2024; Pang et al., 2024; Kim et al., 2023b), but crucially restrict the bottleneck layer to image-level attributes, precluding the use of features that are spatially-localized. At the cost of additional annotation, this two-stage design

improves interpretability and enables test-time human intervention by allowing edits to predicted concepts. Extensions to CBMs include Label-Free CBMs (Oikarinen et al., 2023), Concept Embedding Models (Espinosa Zarlenga et al., 2022), Graph CBMs (Xu et al., 2025), probabilistic CBMs (Kim et al., 2023a), and works that integrate language models or VLMs to generate or align concepts (Yang et al., 2023; Oikarinen et al., 2023; Yan et al., 2023; Hsu et al., 2025; Srivastava et al., 2024; Liu et al., 2024; Gao et al., 2024; Patrício et al., 2025).

More recently, neuro-symbolic approaches have demonstrated potential in grounding AI models in symbols, where a symbol can be represented by a tool (Hsu et al., 2023). One line of work explores programmatic execution of tools for complex visual understanding (Surís et al., 2023; Gupta and Kembhavi, 2023). These approaches leverage a toolbox of pretrained (image-level or pixel-level) tools (e.g., segmentation, detection, object-level tools) and generate code that calls these tools, which is subsequently executed to make a prediction. Besides outperforming non-symbolic visual understanding approaches, these works can be seen as grounding the predictor via the tools that are by themselves interpretable (Hu et al., 2024; Lu et al., 2025).

Our work builds on both CBMs and neuro-symbolic approaches. Instead of a bottleneck layer composed of scalar concepts, our TBM formulation generalizes such works to handle structured features, such as pixel-level features that encode spatially localized information, which is essential in medical imaging. Instead of neuro-symbolic approaches that generate text for composing tool outputs, our TBM can be viewed as a learned tool composition mechanism, which is better equipped to fuse fine-grained, spatially localized features common in medical imaging.

## 3. Method

We are interested in the medical image understanding task, where the input is an image $x$, and the goal is to predict an image-level label $y$. In this work, we consider two clinical prediction settings: tumorous tissue detection in histopathology patches and cancer prediction in dermatology images.

To adapt the interpretability and grounding benefits of tool-use to medical image understanding, we propose the Tool Bottleneck Framework (TBF), which is composed of three components: (1) A toolbox of pretrained medical imaging tools (Section 3.1), (2) a vision-language model (VLM) that serves as a tool selector (Section 3.2), and (3) a novel Tool Bottleneck Model (TBM) that makes the final prediction from tool outputs (Section 3.3).

Figure 1 provides an overview of this framework, and we describe each component in detail below.

### 3.1. Toolbox of Clinically-Relevant Tools

We assume access to a toolbox of $N$ tools, $\mathcal{T} = \{t_1, \ldots, t_N\}$, where each tool $t_i$ outputs clinically-relevant features for a given input image. For a given image and task of interest, we assume there exists a subset of tools that extracts clinically-relevant information for solving the task; for example, nucleus segmenter and mitotic figure detector in tumorous tissue detection, or lesion segmenter and brown color detector in skin lesion diagnosis.

The toolbox may consist of tools designed for a variety of modalities. Furthermore, each tool $t_i$ may produce an output $z_i = t_i(x)$ of varying structure, including a scalar (e.g., lesion count, predicted probability), a vector (e.g., bounding box coordinates), or a spatial map (e.g., segmentation masks, probability maps). Our goal, given this toolbox, is two-fold: to select the relevant tools for a given task and to fuse their outputs to make a prediction.

In our experiments, we build a toolbox $\mathcal{T}$ composed of $N = 12$ tools for all tasks. For histopathology tools, we use the open-source models from the TIAToolbox (Pocock et al., 2022) computational pathology toolbox, specifically the HoVer-Net (Graham et al., 2019) model, which provides nuclear instance segmentation and classification. We convert these predictions into five spatial feature maps: *bounding boxes*, *centroids*, *nucleus types*, *type probabilities*, and *contours*. For dermatology tools, we consider three categories of tools: lesion segmentation, dermoscopic structure maps, and color-based markers. In total we have 7 tools: *lesion segmenter*, *pigment network*, *negative network*, *streaks*, *milia-like cysts*, *malignant-colored pigment marker*, and *brown pigment marker*. We provide additional details on all tools in Appendix C.

## 3.2. Vision-Language Model as a Tool Selector

We formulate tool selection as selecting the $k$ most relevant tools in the toolbox for a given task and image. We use a medical VLM to perform this tool selection. Specifically, the VLM is prompted with $k$, $x$, the toolbox $\mathcal{T}$, and a natural-language description of the task, and it returns a subset of tools $\mathcal{T}_s \subseteq \mathcal{T}$ that can best solve this particular task.

An example prompt is provided below, and an example VLM-selection output is provided in Figure 1:

```
TOOLBOX = {
  "histo_nuc_centroid":  "Returns each nucleus centroid in
                         [x_center, y_center]",
  "histo_nuc_bbox":      "Returns each nucleus bounding box in
                         [x_top_left, y_top_left, width, height]",
  ...

  "derm_lesion_segmenter":         "Segments lesion ROI",
  "derm_pigment_network":          "Detects reticular pigment
                                    network",
  ...
}

You are a medical expert in {modality}. Select tools for a single
task from a fixed toolbox {TOOLBOX}.
Choices must depend on the task and image evidence.
Choose max {k} tools from the toolbox {TOOLBOX}
that are most relevant for solving the task in each image; no
duplicates.
```

The full prompt is provided in Appendix F. Besides a fixed top-$k$ tool selection, we also experiment with a dynamic variant, and present the results in Appendix D.3.

### 3.3. Tool Bottleneck Model (TBM)

We propose the TBM for making predictions given VLM-selected tools, which may be seen as an extension of the Concept Bottleneck Model (CBM) for pixel-level tool outputs common in medical imaging. A TBM takes as input $x$ and the selected tools $\mathcal{T}_s$, and it outputs the final image-level prediction: $y = \text{TBM}(x, \mathcal{T}_s)$. Like CBM, it does so in two steps. First, it computes the tool bottleneck layer $z$, which is the set of all tool outputs for the given image $x$ across all provided tools. Then, a neural network $f_\theta$ takes in $z$ and outputs the final prediction: $y = f_\theta(z)$. During training, all tools are frozen and only the parameters $\theta$ within the TBM are learned.

We propose a simple yet effective implementation of $f_\theta$ that (1) effectively fuses the pixel-level features across tools and (2) accepts any arbitrary selection of tools $\mathcal{T}_s$. To address (1), we rasterize all $z_i$ as pixel-feature maps of size $(C_i \times H \times W)$, ensuring spatial correspondence across tool outputs. All tool maps are concatenated along the channel dimension to form $z = \text{Concat}(z_1, z_2, ..., z_N) \in \mathbb{R}^{C \times H \times W}$, where $C = \sum_i C_i$. Then, we implement $f_\theta$ as a CNN feature extractor followed by a final fully-connected classifier.

To address (2), we train TBMs with *tool knockout* augmentation (Nguyen et al., 2025). Specifically, given a tool selection $\mathcal{T}_s$, we replace the outputs of non-selected tools $\mathcal{T} \setminus \mathcal{T}_s$ with a fixed placeholder value $\bar{z}_i$ (e.g., a constant map of -1s). Prior work has shown that this knockout strategy is equivalent to an implicit multi-task objective that jointly learns estimators of $y$ conditioned on all tools and its subsets (see Appendix E). Tool knockout also permits a "leave-one-tool-out" analysis, which provides the user a notion of tool importance useful for model interpretability. We demonstrate this in Section 5.3.

In our experiments, we found that random perturbation of tool selection during training improved results. Specifically, instead of directly passing the selected tools $\mathcal{T}_s$ to the TBM, we sample from a Bernoulli distribution with parameter $p = (1 - \alpha)0.5 + \alpha s_i$ for each tool; here, $s_i \in \{0, 1\}$ is a binary selection indicator denoting whether or not the VLM selects tool $i$, and $\alpha \in [0, 1]$ is a hyperparameter that controls the strength of the VLM prior. We ablate this perturbation in our experiments (see also Appendix D.2).

## 4. Experiments

### 4.1. Dataset and Tools

We report results on three medical image understanding tasks (Camelyon17, ISIC-BM, and ISIC-MN) derived from two medical imaging datasets in histopathology and dermatology. Detailed descriptions of each dataset and tool are provided in the Appendix.

The Camelyon17-WILDS dataset (Koh et al., 2021) is adapted from the CAMELYON17 challenge (Litjens et al., 2018), which consists of whole-slide images (WSIs) of breast cancer metastases in lymph node sections. Each WSI is manually annotated by pathologists to mark tumor regions, from which non-overlapping $96 \times 96$ pixel patches are extracted and labeled as either {tumor, normal}.

The ISIC 2017 dataset (Codella et al., 2018) provides 2000 dermoscopic images for skin lesion analysis, with three primary diagnostic categories: melanoma (malignant, melanocytic), nevus (benign, melanocytic), and seborrheic keratosis (SK) (benign, non-melanocytic). We evaluate two clinically-relevant binary classification tasks: 1) ISIC-BM: Benign vs. Malig-

nant, where melanoma is treated as malignant and {nevus, SK} as benign; and 2) ISIC-MN: Melanocytic vs. Non-melanocytic, where {melanoma, nevus} are grouped as melanocytic and SK as non-melanocytic. The validation and test sets contain 150 and 600 images for ISIC-BM and ISIC-MN, respectively.

## 4.2. Experimental Setup

We use MedGemma (Sellergren et al., 2025) as the tool selector for TBF and set $k = 3$ for all experiments. We set $\alpha = 0.9$ for Camelyon17 and ISIC-BM, and $\alpha = 0.8$ for ISIC-MN (See Section 3.3). These values were tuned via grid search over $k \in \{2, 3, 4\}$ and $\alpha \in \{0.5, 0.6, 0.7, 0.8, 0.9\}$ (Appendix D.2) with respect to their validation sets.

For Camelyon17, we subsample 5,000 patches to reduce computational cost while retaining patient-level diversity, drawing 100 patches from each patient (WSI) folder following the official WILDS splits. All models are trained on the training split and evaluated on the held-out in-distribution (ID) and out-of-distribution (OOD) hospital splits. We use EfficientNet-B0 for $f_\theta$ pretrained on ImageNet, modified to accept $C$ input channels. All Camelyon TBMs are optimized with cross-entropy loss using Adam with a learning rate of $10^{-4}$ and weight decay of $10^{-4}$, for 40 epochs. We select the best checkpoint based on validation performance on the corresponding ID Val split. Due to the Camelyon dataset being evenly balanced across classes and following prior work (Koh et al., 2021), we report accuracy in all results.

For ISIC, we train TBMs on both binary tasks (ISIC-BM and ISIC-MN). Images are resized to a fixed input resolution of $224 \times 224$ and augmented with random flips. For $f_\theta$, we use a CNN with 4 convolutional blocks (32, 32, 64, 128 channels) trained from scratch on the ISIC training set. Both ISIC tasks exhibit class imbalance and we train with a class-weighted binary cross-entropy loss to handle this. All ISIC TBMs are trained using the Adam optimizer (learning rate $1 \times 10^{-3}$, weight decay $1 \times 10^{-4}$) with a cosine annealing schedule over 20 epochs. For both ISIC-BM and ISIC-MN, we report area-under-the-receiver-operating-curve (AUC) in all results. For all tasks, tool outputs are rasterized into multi-channel maps whose values lie in $\{0, 1\}$ and concatenated along the channel dimension. Further details are in Appendix C. For all experiments, we set $\bar{z}_i = -\mathbf{1}^{C_i \times H \times W}$, i.e. a constant map of -1s.

## 4.3. Baselines

We compare against two classes of baselines. The first class consists of state-of-the-art, zero-shot VLMs with and without tool-use. For all tool-use frameworks, we implement them such that they use our toolbox $\mathcal{T}$. *MedGemma* (Sellergren et al., 2025) is a closed-source, medical-VLM; it does not expose a way to integrate custom tools. As such, we include tool outputs as additional prompts to the model, and denote this baseline as *MedGemma w/ Tool Prompts*. We additionally evaluate Gemma 3 (Kamath et al., 2025) zero-shot and with tool outputs as additional prompts (denoted as *Gemma w/ Tool Prompts*). These additional prompts serve as pixel-level clinical priors to control for differences in pixel-level supervision and inductive biases that may not be accounted for in other baselines. *VisProg* (Gupta and Kembhavi, 2023) is a neuro-symbolic method for image understanding that uses a pretrained VLM to write executable code consisting of tool calls for reasoning

Table 1: TBF performance against baselines and ablations across Camelyon17 (Accuracy), ISIC-BM (AUC), ISIC-MN (AUC). Top group: zero-shot baselines with and without tools. Middle group: trained/fine-tuned baselines. Bottom group: our proposed framework and ablations. **Bolded** is best and underlined is second best. For ablations, the delta is computed with respect to TBF (ours).

| Model | Camelyon17 | ISIC-BM | ISIC-MN |
|---|---|---|---|
| Gemma (Kamath et al., 2025) | 50.0 | 49.5 | 50.4 |
| Gemma w/ Tool Prompts | 50.9 | 47.1 | 48.8 |
| MedGemma (Sellergren et al., 2025) | 50.0 | 44.4 | 48.7 |
| MedGemma w/ Tool Prompts | 50.0 | 46.8 | 50.0 |
| VisProg (Gupta and Kembhavi, 2023) | 50.4 | 50.0 | 50.0 |
| LlavaMed (Li et al., 2023) | 50.0 | 49.4 | 50.0 |
| EfficientNet (Tan and Le, 2019) | 88.6 | **78.4** | 91.2 |
| Y-Net (Mehta et al., 2018) | 88.2 | 65.8 | 86.6 |
| LlavaMed FT | 66.2 | 51.5 | 58.0 |
| TBF (ours) | **92.3** | 77.5 | **91.7** |
| → without perturbation ($\alpha = 1$) | -2.1 | -2.1 | -1.4 |
| → with all modality-specific tools | -0.2 | -2.2 | -0.7 |

over visual inputs. *LLaVA-Med* (Li et al., 2023) is an open-source, medical-VLM. We first evaluate it in a zero-shot setting.

The second class consists of models trained or fine-tuned on our data. *EfficientNet* (Tan and Le, 2019) serves as a standard black-box CNN baseline trained directly on resized raw images without any tool inputs. *Y-Net* (Mehta et al., 2018) is an extension of U-Net (Ronneberger et al., 2015) originally proposed for joint segmentation and classification, where a pixel-level loss is applied at the final layer of the U-Net and an image-level loss is applied at the middle bottleneck layer of the U-Net. We train Y-Net to match the amount and types of data as our proposed model; specifically, pixel-level tool outputs are optimized at the final layer, while an image-level classification loss is minimized at the middle layer. Finally, we include a finetuned *LLaVA-Med (FT)* baseline, where the pretrained LlaVA-Med backbone is finetuned on data. More details are provided in Appendix B.

## 5. Results

### 5.1. Main Results

We compare the performance of TBF and its variants against all baselines for Camelyon17, ISIC-BM, and ISIC-MN in Table 1.

Generally, we observe that TBF outperforms or is on-par with baselines across all tasks we tested. TBF is the best-performing model on Camelyon17 and ISIC-MN and is second-best for ISIC-BM, where EfficientNet performs the best. Notably, TBF outperforms Y-Net,

despite both being trained on the same amount and types of image-level and pixel-level data. We attribute this to the TBF formulation – since $f_\theta$ only takes as input VLM-selected, clinically-relevant features, we hypothesize that learning predictors on these features (instead of raw images) induces more robust predictions. Additionally, MedGemma, MedGemma w/ Tool Prompts, and VisProg both perform poorly (VisProg gives constant answers, resulting in an AUC of 0.5). Since all models can use the same toolbox $\mathcal{T}$ but compose them through zero-shot, text-based mechanisms, this suggests the importance of tool-use frameworks that use learned composition mechanisms for best performance in medical imaging settings.

The bottom of Table 1 depicts two ablation experiments. First, we ablate the perturbation of VLM tool selections described in Section 3.3; this corresponds to using a value of $\alpha = 1$. Across all tasks, we observe a 1-2% drop in performance without random perturbation of VLM-selected tools. We hypothesize that perturbed VLM tool selection increases robustness of the TBM, since it sees a wider variety of tool combinations at training compared to without random perturbation and/or without tool selection. Second, we ablate the VLM tool selector and simply pass in all modality-specific tools. Note that in the case of large $N$, this is computationally intractable. We observe that the TBM performs slightly worse when using all modality-specific tools than when using VLM tool selection.

We refer the reader to Appendix D.1 for results using non-pretrained models.

## 5.2. Data Efficiency

We hypothesize that TBF has advantages in low-data regimes, since the architectural design of the TBF encodes clinically-relevant inductive biases. To assess this, we conduct data-efficiency experiments on both Camelyon17 and ISIC-MN (Figure 2). For each dataset, we train TBM and EfficientNet on increasingly larger subsets of the training set. The subsets are chosen at random while ensuring that classes are balanced.

Across all training-set sizes, TBF outperforms EfficientNet, with especially large gains in the small-data regime (4–64 images). For example, with only four labeled examples of Camelyon17 images, TBF reaches ∼0.64 accuracy while the black-box baseline reaches only ∼0.57. At larger training sizes, we also observe that TBF exhibits lower variance across seeds, which we attribute to the reduced hypothesis space of $f_\theta$ due to its clinical grounding.

## 5.3. Analysis

We are interested in analyzing the "importance" of each tool for a given task and how that relates to the distribution of VLM tool selections during training. To measure importance of a given tool, we knockout that tool while passing all other modality-specific tools, and compute the resulting change in performance averaged across the validation set. Specifically, the importance of tool $t_i$ is defined as:

$$\mathcal{I}(t_i) = \frac{1}{|\mathcal{D}_v|} \sum_{(\boldsymbol{x}, \boldsymbol{y}) \in \mathcal{D}_v} m\left(\text{TBM}(\boldsymbol{x}, \mathcal{T}), \boldsymbol{y}\right) - m\left(\text{TBM}(\boldsymbol{x}, \mathcal{T}_{-\{t_i\}}), \boldsymbol{y}\right), \tag{1}$$

where $\mathcal{D}_v$ is the validation set, $m$ is the performance metric, and $\mathcal{T}_{-\{t_i\}}$ denotes the toolbox with $t_i$ dropped. This is closely related to the notion of influence functions (Koh and Liang, 2017).

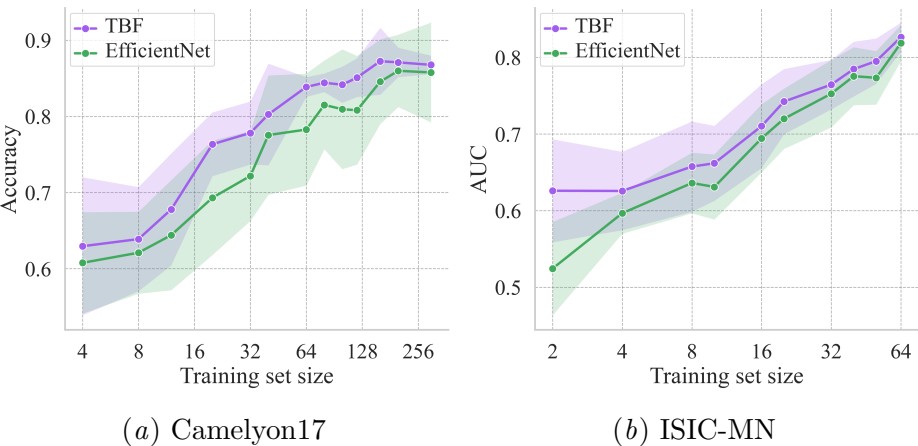

(a) Camelyon17           (b) ISIC-MN

Figure 2: Model performance of TBF vs. EfficientNet baseline over varying training set sizes in log scale. Mean ± 95% CI over seeds. TBF exhibits improved performance across all training set sizes.

We perform this analysis on Camelyon17 and ISIC-MN tasks (Fig. 3), where $m$ is accuracy and AUC, respectively. Alongside tool importance, we also plot the normalized frequency of tool selections across the training set, since tool importance may be correlated with how frequently that tool is selected by the VLM during training.

We observe that for Camelyon, the nucleus contour tool has the highest importance, despite the VLM selecting the contour tool relatively less than nucleus bbox and centroid tools. Similarly, for ISIC, we observe that the lesion segmentation tool, pigment network tool, and brown marker tool have the highest importance. This pattern is consistent with literature, where, for instance, nuclei count and irregularity are known to be correlated with pathology (Kuenen-Boumeester et al., 1984). Similarly, pigment networks and border irregularity (extracted by lesion segmentations) are important in diagnosing malignant melanoma (Anantha et al., 2004; Stolz, 1994).

In Appendix D.4, we experiment with directly intervening on tool outputs as another method to probe TBF's decision-making. In Appendix D.5, we visualize all training-time tool combinations, not just overall selection frequency.

## 6. Conclusion and Future Work

We presented the Tool Bottleneck Framework (TBF), a tool-use framework for clinically-informed and interpretable medical image understanding. TBF leverages a VLM to select tools from a pre-specified toolbox most relevant for the task at hand. Unlike prior works that compose these tools via text, our framework composed selected tools using a learned Tool Bottleneck Model (TBM), which computes the tool outputs on the given image and fuses the tool outputs to make a prediction. We present a simple yet effective strategy of training TBMs such that they accept any arbitrary subset of tools via tool knockout

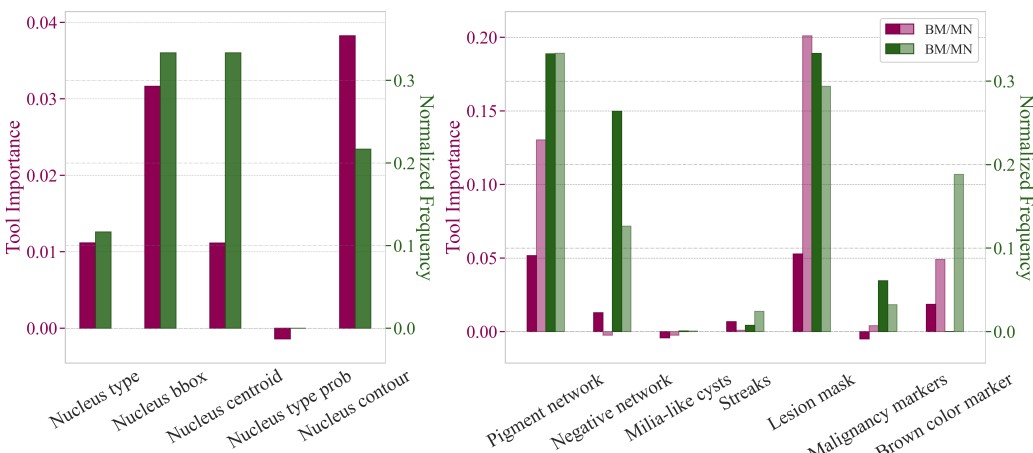

Figure 3: Tool-wise importance (Eq. 1) and normalized frequency of VLM tool selections for TBM across Camelyon17 **(left)** and ISIC-BM/-MN **(right)**. In each plot, the left axis shows the relative importance of each tool measured by the change in Accuracy (Camelyon17) and AUC (ISIC) when tools are individually removed during inference. The right axis shows the normalized frequency of tools selected by MedGemma during training.

augmentation. On tasks derived from histopathology and dermatology, we observe that TBF outperforms state-of-the-art tool-use frameworks while being more interpretable and data-efficient as compared to CNNs and VLMs. Additionally, we propose a way to interrogate tool importance for further interpretability. As a future direction, scaling our work to use a more comprehensive set of tools would yield a more general framework for medical image understanding. In addition, optimizing the tool selection VLM would also be an interesting direction for exploration.

## Acknowledgments

This work was partially funded by the NIH Grant R01AG089169, R33AG084471, Stanford HAI Hoffman-Yee Award, NSF RI #2211258, and Stanford HAI postdoctoral fellowship.

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

# Appendix A. Dataset Details

- The **Camelyon17-WILDS** dataset (Koh et al., 2021) is adapted from the CAME-LYON17 challenge (Litjens et al., 2018), which consists of whole-slide images (WSIs) of breast cancer metastases in lymph node sections. Each WSI is manually annotated by pathologists to mark tumor regions, from which non-overlapping $96 \times 96$ pixel patches are extracted and labeled as either *Tumor* or *Normal*. A patch is labeled *Tumor* if the central $32 \times 32$ region contains any tumor tissue, and *Normal* if it contains no tumor and at least 20% normal tissue in that central region.

  The dataset comprises approximately 450,000 patches extracted from 50 WSIs of breast cancer metastases in lymph node sections, collected from five hospitals in the Netherlands (10 WSIs per hospital). Each WSI was manually annotated by expert pathologists, and the corresponding segmentation masks were used to assign patch-level labels. Metadata includes the slide ID (WSI) and the hospital identifier (domain) for each patch.

  To evaluate cross-domain generalization, data is split by hospital as follows: the *training* split contains 302,436 patches from 30 WSIs (10 WSIs each from 3 hospitals): the *validation (OOD)* split contains 34,904 patches from 10 WSIs from the 4th hospital; the *test (OOD)* split contains 85,054 patches from 10 WSIs from the 5th hospital, chosen for its distinct staining characteristics and visual style; and the *validation (ID)* split contains 33,560 patches from the same 30 WSIs used for training. This setup ensures that no WSIs or hospitals overlap between the training and OOD splits, making the benchmark well-suited for studying domain generalization and robustness to inter-hospital variability. The task we consider is binary classification: given a $96 \times 96$ histopathology patch, predict whether the central $32 \times 32$ region contains any tumor tissue.

- The **ISIC 2017** dataset provides dermoscopic images for skin lesion analysis, with three primary diagnostic categories: melanoma (malignant, melanocytic), nevus (benign, melanocytic), and seborrheic keratosis (SK) (benign, non-melanocytic). We evaluate two clinically relevant binary classification tasks: (1) *Benign vs. Malignant*, where melanoma is treated as malignant and {nevus, SK} as benign; and (2) *Melanocytic vs. Non-melanocytic*, where {melanoma, nevus} are grouped as melanocytic and SK as non-melanocytic.

  The official training set contains 2,000 JPEG dermoscopic images with ground-truth diagnoses and a CSV of minimal clinical metadata (image_id, age_approximate, sex). Class counts in the training set are 374 melanoma, 254 SK, and 1,372 nevi. The validation and test sets contain 150 and 600 images, respectively. Ground-truth labels are provided via two binary indicators: a melanoma indicator (1 for melanoma, 0 otherwise) and an SK indicator (1 for SK, 0 otherwise), which we map to the two binary tasks above. Images are high-resolution dermoscopic photographs ($767 \times 1022$ pixels), which we preprocess to a fixed input resolution before tool extraction and model training.

  For the malignant vs. benign task, the malignant class is underrepresented (Train: 374/2000; Val: 30/150; Test: 117/600), while for the melanocytic vs. non-melanocytic

task the non-melanocytic (SK) class is the minority (Train: 254/2000; Val: 42/150; Test: 90/600). We compute a positive-class weight from the training distribution and apply it in the BCE loss for each task.

## Appendix B. Baseline Details

- **MedGemma Zero-Shot.** We evaluate MedGemma in a zero-shot setting using text prompts. We prompt MedGemma to return the predicted binary label, as well as a probability score $[0, 1]$ for each prediction, in order to report Accuracy and AUC. This baseline represents the performance of a general-purpose medical VLM. Similar to TBF, it can integrate domain knowledge and operate across modalities. However, unlike TBF, it is not explicitly interpretable or trainable, as its reasoning process is opaque and not intervenable, and its outputs cannot be decomposed into verifiable, tool-level predictions.

- **MedGemma w/ Tool Prompts**. We evaluate a tool-use variant of MedGemma that operates in a two-turn, zero-shot fashion using our toolbox $\mathcal{T}$. In the first turn, we prompt MedGemma as a tool selector detailed in Section 3.2, with the full prompts listed in Appendix D.3. The outputs of the selected tools are then rasterized into spatial map images and then passed as input to MedGemma in a second turn as additional visual inputs, along with the original image and task to obtain the final predicted label probability scores. We do not fine-tune the model on any task labels. This baseline represents a medical VLM that can both select and utilize the same tools used by TBF.

- **Gemma Zero-Shot.** We evaluate Gemma 3 in the same way as MedGemma Zero-Shot. This baseline represents the performance of a general VLM.

- **Gemma w/ Tool Prompts**. We evaluate a tool-use variant of Gemma 3 in the same way as MedGemma w/ Tool Prompts. This baseline represents a general VLM that can both select and utilize the same tools used by TBF.

- **VisProg**. We evaluate VisProg (Gupta and Kembhavi, 2023), a tool-use system that answers visual questions by composing executable programs, which access tools from our toolbox $\mathcal{T}$. For all tasks, we enable VisProg to utilize the tools specific for the given modality. We prompt it to return both a binary classification as well as a predicted probability to compute Accuracy and AUC. However, we found that VisProg fails to give outputs that are not binary, even when specifically prompted to do so. Thus, to compute AUC, we map the binary prediction to $\{0, 1\}$ to obtain probabilities. For both datasets, we do not fine-tune the code generation VLM or the VQA (BLIP) VLM on our data. This baseline therefore reflects the performance of a code-generation tool-use model that performs text-based compositional reasoning while explicitly leveraging the same toolbox used by our TBF.

- **EfficientNet.** For comparison with a standard CNN, we use the EfficientNet-B0 architecture (Tan and Le, 2019), which is a black-box baseline trained directly on raw images, without any tool inputs. Images are resized to match the TBF input

resolution ($96 \times 96$ for Camelyon17 and $224 \times 224$ for ISIC), and we use the same optimizer configuration as for TBF.

- **Y-Net.** TBF leverages additional data in the form of tools and their pixel-level outputs. To disentangle the effect of our model formulation and increased amount and type of data, we compare against a popular Y-Net–style segmentation-for-classification model (Mehta et al., 2018). Y-Net leverages pixel-wise supervision in addition to image-level labels, but does not use explicit tool decomposition. Thus, Y-Net and TBF are trained on the same amount and type of data. Training is performed in two stages: In Stage 1, we train Y-Net with only the segmentation head enabled to predict tumor-versus-background masks from raw patches ($96 \times 96$ for Camelyon17 and $224 \times 224$ for ISIC), using tumor masks, derived from HoverNet outputs for Camelyon and lesion segmentation masks for ISIC, as the supervision target. The model is optimized with pixel-level cross-entropy loss using Adam (learning rate $10^{-3}$, batch size 16) for 25 epochs, and monitored with Dice score. In Stage 2, we jointly train the segmentation and classification head with the combined loss $\mathcal{L} = \mathcal{L}_{\text{seg}} + \mathcal{L}_{\text{cls}}$, where both $\mathcal{L}_{\text{seg}}$ and $\mathcal{L}_{\text{cls}}$ are cross-entropy losses for the segmentation mask and binary label, respectively. Stage 2 uses Adam with a lower learning rate ($3 \times 10^{-4}$), batch size 16, and 20 epochs. For both datasets, Y-Net is trained on the same data splits and training setup as TBF.

- **LLaVA-Med Zero-Shot and Finetuned.** We use the released LLaVA-Med (Li et al., 2023) `llava-med-v1.5-mistral-7b` checkpoint for all LLaVA-Med experiments. For the zero-shot setting, we keep all model weights frozen and only prompt the model with task-specific instructions and answer formats. The exact prompts for all tasks are provided in Appendix F.1. We ask the model to output a scalar confidence score in $[0, 1]$ for the positive class and the binary class label. We use the continuous scores to compute AUC. For the finetuned setting (LLaVA-Med FT), we further finetune on the training set, where the corresponding text for each image is {"tumor", "no tumor"} for Camelyon17, {"malignant", "benign"} for ISIC-BM, and {"melanocytic", "non-melanocytic"} for ISIC-MN. We unfreeze the last two vision transformer blocks and apply a LoRA adapter to the language model's attention and MLP projection layers. We train for 4 epochs with AdamW, with learning rate $5 \times 10^{-6}$, weight decay 0.01, and a cosine learning-rate schedule with a warmup ratio of 0.1.

## Appendix C. Tool Details

### C.1. Camelyon17

For Camelyon17, we construct histopathology tools by using the open-source TIAToolbox (Pocock et al., 2022) computational pathology toolbox that provides end-to-end pathology image analysis. We specifically use the HoVer-Net (Graham et al., 2019) model for nucleus-level predictions, specifically segmentation and classification. We apply this tool to each Camelyon17 patch, producing instance-level nuclei predictions, including per-nucleus class type labels, class type label probabilities, bounding boxes, segmentation contours, and

centroid coordinates. For each patch, HoverNet returns a dictionary indexed by a unique nucleus identifier `nuc_id`, where every entry contains: `box`, `centroid`, `contour`, `prob`, and `type`, corresponding to the nuclei instance-level predictions of bounding boxes, centroid coordinates, segmentation contours, class type label probabilities, and class type labels. We convert these instance-level outputs into spatial feature maps rasterized onto a $96 \times 96$ grid aligned with the corresponding tissue patch:

- **Bounding boxes (`histo_nuc_bbox`).** The `box` field stores bounding box coordinates in the format $[x_{\text{top-left}}, y_{\text{top-left}}, \text{width}, \text{height}]$. For each nucleus, we fill its bounding box on a binary canvas and then downsample to $96 \times 96$. Overlapping boxes are merged by taking the per-pixel maximum, yielding a single-channel map that highlights regions with dense or enlarged nuclei.

- **Centroids (`histo_nuc_centroid`).** The `centroid` field stores the nucleus center in $[x_{\text{centre}}, y_{\text{centre}}]$ coordinates. We place a small blob at each centroid location on a blank canvas and resample to $96 \times 96$, producing a single-channel map that encodes the spatial distribution of nuclei (nuclear density and clustering).

- **Contours (`histo_nuc_contour`).** The `contour` field is a list of points forming the polygonal boundary of each nucleus. We rasterize these polygons by drawing only the boundary pixels on a binary canvas and downsampling to $96 \times 96$. This single-channel map emphasizes nuclear shape and boundary irregularity while suppressing interior regions.

- **Nucleus types (`histo_nuc_type`).** The `type` field stores the predicted discrete class label for each nucleus as an integer in $\{0, \ldots, 5\}$, corresponding to categories such as background, neoplastic epithelial, inflammatory, connective, dead, and non-neoplastic epithelial cells. We create a multi-channel one-hot tensor by assigning each pixel within a nucleus to its predicted class and stacking the resulting binary masks across channels. This yields a $C_{\text{type}} \times 96 \times 96$ tensor (with $C_{\text{type}} = 6$ in our implementation) that captures spatial distributions of different cell types.

- **Type probabilities (`histo_nuc_type_prob`).** The `prob` field stores the confidence (probability) of the predicted type for each nucleus. We propagate this scalar confidence value to all pixels in the corresponding nucleus mask and aggregate overlapping instances by taking the per-pixel maximum. The result is a single-channel confidence map in $[0, 1]$ that highlights regions where the model is confident about nuclear identity versus uncertain or ambiguous regions.

All five histopathology tools are computed once using the pretrained TIAToolbox HoverNet model and kept fixed. We do not fine-tune the underlying nucleus segmentation tool on Camelyon17. The resulting maps are concatenated along the channel dimension into a tool tensor of shape $(C_{\text{tool}}, 96, 96)$, with $C_{\text{tool}} = C_{\text{type}} + 4$. As described in Section 4.1, all Camelyon17 tool maps are scaled to lie in $[0, 1]$ before masking; dropped tools are represented by channels filled with a constant "missing" value of $-1$ (a constant map of $-1$s) and are handled consistently across all masking regimes. Details on filling dropped tools with the "missing" values are provided in Section E.

## C.2. ISIC

For ISIC 2017, images are high-resolution dermoscopic photographs ($767 \times 1022$ pixels), which we preprocess to a fixed input resolution before tool extraction. We construct a dermatology toolbox with seven tools: lesion segmentation, pigment network, negative network, streaks, milia-like cysts, malignant-color pigment marker, brown pigment marker. Each tool produces a single-channel spatial map that is rasterized to a common $H \times W$ resolution ($224 \times 224$ in our experiments) and stacked along the channel dimension:

- **Lesion segmentation (`derm_lesion_segmenter`).** First, we train our own lightweight U-Net–style lesion segmentation model using the pixel-wise lesion masks provided in the ISIC 2017 training set. The model takes the raw dermoscopic RGB image as input and outputs a high-resolution binary lesion mask, which we treat as a spatial tool map aligned with the original image. This mask is used both as a standalone tool channel and as a region-of-interest mask for the color-based marker tools described below.

- **Dermoscopic structure maps (`derm_pigment_network, derm_negative_network, derm_streaks_detector, derm_milia_like_cyst_detector`).** To obtain predicted dermoscopic feature maps, we integrate the open-source skinisic model (Kawahara and Hamarneh, 2019), a VGG16-based fully convolutional network finetuned on ISIC superpixel labels converted into per-pixel multi-channel segmentations, which predicts pixel-wise probability maps for pigment network, negative network, streaks, and milia-like cysts. We pass each ISIC 2017 image through skinisic to obtain four probability maps, which we threshold (0.5 by default) to obtain binary pixel maps that are resized to $H \times W$ and passed into the TBM as tool outputs.

- **Color-based marker tools (`derm_marker_malignant_union, derm_marker_browns`).** Finally, we include two additional color-based tools inspired by prior work on color and texture features for dermoscopic lesion classification (Marques et al., 2012) and color-constancy-based preprocessing for robust skin image analysis (Ciurea and Funt, 2003). Following these approaches, we first apply a simple "shades of gray" color constancy transform (Ciurea and Funt, 2003) to the RGB image to reduce illumination variability. In this normalized color space, we apply handcrafted threshold rules over the $(R, G, B)$ channels to detect canonical melanoma-related color patterns (e.g., very dark/black regions, blue-gray areas, white structures) and different shades of brown. Small connected components and holes below a minimum area threshold are removed via morphological post-processing, and all marker maps are restricted to the predicted lesion region by intersecting with the lesion segmentation mask. We then aggregate these fine-grained markers into two compact binary tools: (i) a *malignant-colored pigment marker*, defined as the union of black, blue-gray, and white melanoma-associated colors, and (ii) a *brown pigment marker*, defined as the union of light- and dark-brown regions. Both maps are rasterized as $H \times W$ binary images aligned with the dermoscopic image and appended as additional channels in the tool stack.

All seven ISIC tools are computed once per image and then kept fixed. The resulting maps are concatenated along the channel dimension into a tool tensor of shape $(C_{\text{tool}}, 224, 224)$ (with $C_{\text{tool}} = 7$ in our main experiments). As described in Section 4.1, all ISIC tool channels

Table 2: Accuracy (%) on Camelyon17 for TBF variants and baselines with and without ImageNet pretrained weights. Best value per pretraining type is **bolded**.

| Model Variant | ImageNet pretrained? | Acc (%) |
|---|---|---|
| EfficientNet | ✗ | 72.6 |
| Y-Net | ✗ | 72.0 |
| TBF (ours) | ✗ | **86.7** |
| TBF without perturbation ($\alpha =1$) | ✗ | 86.5 |
| EfficientNet | ✓ | 88.6 |
| Y-Net | ✓ | 88.2 |
| TBF (ours) | ✓ | **92.3** |
| TBF without perturbation ($\alpha =1$) | ✓ | 90.2 |

are scaled to lie in $[0, 1]$ before masking. Similar to Camelyon17, we represent dropped tools with a constant map of $-1$s.

## Appendix D. Additional Results

### D.1. Effect of Pretraining

Table 2 summarizes TBF performance under scratch vs. ImageNet initialization when evaluated on Camelyon17. TBF achieves 86.7% accuracy when trained from scratch, and increases to 92.3%. Even without pretraining, TBF already outperforms non-pretrained EfficientNet and Y-Net baselines (72.6% and 72.0%), indicating that the tool bottleneck structure itself provides generalization benefits under limited supervision.

Among pretrained models, TBF with and without perturbation exceed the accuracy of the pretrained EfficientNet and Y-Net (88.6% and 88.2%). This demonstrates that TBF not only benefits from better representations, but also from decomposing the image into domain-grounded tool features, reducing reliance on spurious dataset-specific effects and improving robustness under distribution shift.

### D.2. Tool Sampling Strategies

In addition to TBF, TBF without perturbation ($\alpha = 1$), and TBF with all modality-specific tools described in Table 1, we explore other tool sampling strategies. We explore (1) *Bernoulli*: sampling from a Bernoulli independently for each tool, (2) *Random top-k*: fixing $k$ tools per image by randomly selecting $k$ tools for each image, and (3) $\alpha = c$: sweeping $\alpha$ values over varying values of $c$ to control the strength of the VLM prior and tool perturbation, as described in Section 3.3. We set $k = 3$ for all experiments. TBF ablation results are shown in Table 3.

Table 3: Performance of TBF ablations across Camelyon17 (Accuracy) and ISIC 2017 (AUC).

| Model | Camelyon17 | ISIC-BM | ISIC-MN |
|---|---|---|---|
| Bernoulli | 90.3 | 72.7 | 90.1 |
| $\alpha = 0.5$ | 90.5 | 75.3 | 88.9 |
| $\alpha = 0.6$ | 91.3 | 75.1 | 89.1 |
| $\alpha = 0.7$ | 90.7 | 75.6 | 91.2 |
| $\alpha = 0.8$ | 91.6 | 75.3 | 91.9 |
| $\alpha = 0.9$ | **92.3** | **77.0** | 90.6 |
| $\alpha = 1.0$ | 90.2 | 74.9 | 90.4 |
| Random top-$k$ | 91.7 | 76.1 | **92.0** |
| Dynamic top-$k$ | 91.4 | 66.5 | 90.4 |
| All modality-specific tools | 92.1 | 74.8 | 91.1 |

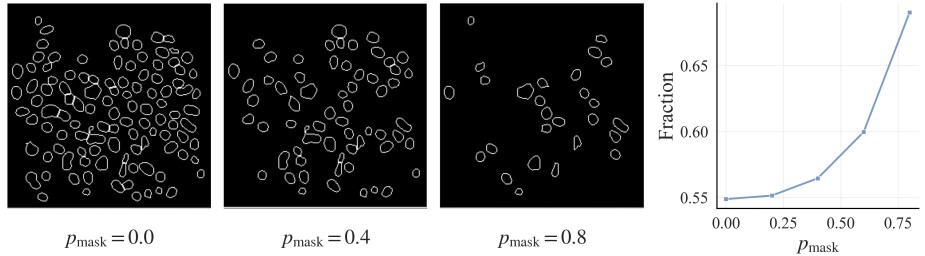

Figure 4: **(a)-(e)**: Visualization of the nuclei–dropout intervention on two example Camelyon17 contour maps. For each example patch, we randomly remove individual nuclei by masking them out with probability $p_{\text{mask}}$, which we sweep across $p_{\text{mask}} \in \{0.0, 0.2, 0.4, 0.6, 0.8\}$ (from left to right) to randomly mask out nuclei in the tool output maps. **(f)**: As $p_{\text{mask}}$ is increased (dropout increased), the fraction of images with a *Normal* label prediction increases monotonically with dropout.

### D.3. VLM Prompting Strategies

In addition to fixed top-$k$, we also experiment with dynamic top-$k$, which permits a dynamic number of tools per image but maintains $k$ tools per image on average over the training set. Specifically, instead of prompting the VLM to output a fixed selection of $k$ tools per image, we prompt the VLM to score each tool between $[0, 1]$. Then, we collect all tool scores for all images in the training set, and compute the cutoff score that corresponds to $k$ tools per image on average. This cutoff is used for all dataset splits. The results are shown in Table 3.

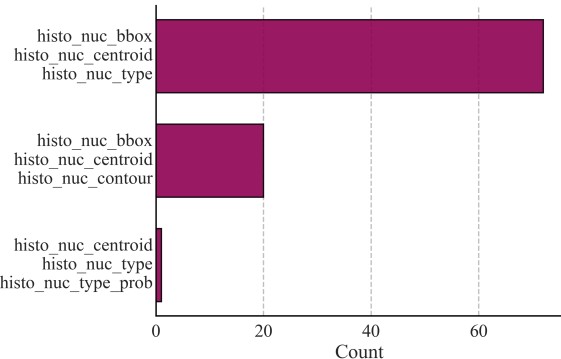

Figure 5: Distribution of MedGemma selected tool combinations for Camelyon17

### D.4. Tool Output Intervention

Besides tool importance discussed in Section 5.3, another method for interrogating TBF's decision-making is to intervene or manipulate the tool outputs, since the features encoded in the tool outputs are clinically meaningful. To test whether TBF relies on nuclei-related features in a clinically meaningful way, we perform a nuclei–dropout intervention on Camelyon17. For each patch, we drop individual nucleus instances independently with probability $p_{\text{mask}}$. When a nucleus is dropped, we set its corresponding tool maps (centroid, bounding-box fill, contour, type one-hot, and type probability) to the background value at the associated pixels (See Figure 4).

We sweep $p_{\text{mask}} \in \{0.0, 0.2, 0.4, 0.6, 0.8\}$ over all nuclei in the validation set, and compute the fraction of patches predicted as Normal, $\Pr(\hat{y} = \text{Normal})$. We observe that as $p_{\text{mask}}$ increases, $P(\hat{y} = \text{Normal})$ increases. This behavior is consistent with established histopathology criteria, where higher nuclear density is a characteristic of malignant breast lesions and correlate with worse prognosis(Narasimha et al., 2013). Nuclear morphology studies report that increased nuclear area is associated with node-positive and higher-grade breast carcinomas (Kuenen-Boumeester et al., 1984; Pienta and Coffey, 1991).

### D.5. Combinations of VLM Tool Selections

In addition to the normalized frequency of tool selections in training depicted in Fig. 3, we also depict the distribution of unique combinations of MedGemma selected tools (top 3) for Camelyon17 and ISIC in Figures 5 and 6.

## Appendix E. Details on Tool Knockout

We show that our tool knockout augmentation enables $f_\theta$ to estimate the full conditional (the distribution of $Y = y$ conditioned on all tool outputs) and all marginal conditionals (the distribution $Y = y$ conditioned on any subset of tool outputs). Our argument follows the theoretical analysis of Knockout by Nguyen et al. (Nguyen et al., 2025).

Let $\boldsymbol{z}_i = t_i(\boldsymbol{x})$ denote the tool output for the $i$'th tool. Let $\boldsymbol{Z} = (\boldsymbol{z}_1, \ldots, \boldsymbol{z}_N)$ denote the collection of all $N$ tool outputs. Let $\mathcal{M}$ denote an indicator set of knocked-out tools. Let $\boldsymbol{M}$ denote the corresponding binary mask vector $\boldsymbol{M} = (M_1, \ldots, M_N) \in \{0, 1\}^N$. Note

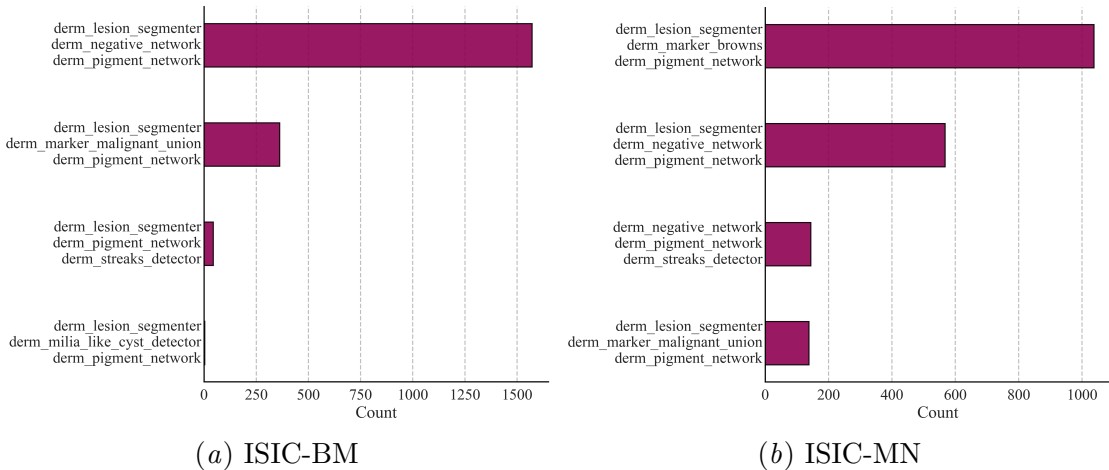

$(a)$ ISIC-BM          $(b)$ ISIC-MN

Figure 6: Distribution of MedGemma selected tools combinations for top 3 tools across the ISIC 2017 tasks

that $M$ is sampled independently of $(Z, Y)$. $Z_{\mathcal{M}}$ denotes $Z$ with elements knocked-out according to indices in $\mathcal{M}$, and $Z_{-\mathcal{M}}$ denotes all non-knocked-out elements. We denote the full conditional as $p(Y \mid Z)$ and all marginal conditionals as $p(Y \mid Z_{-\mathcal{M}})$.

During training, we construct knockout-augmented inputs

$$Z'(M, Z) = M \odot \bar{z} + (1 - M) \odot Z,$$

where $\bar{z}$ is a placeholder feature map chosen to lie outside the support (or in a negligible-density region) of $Z$ and $\odot$ is element-wise multiplication.[1] This ensures the equivalence

$$Z'_{\mathcal{M}} = \bar{z}_{\mathcal{M}} \iff M_{\mathcal{M}} = 1, \qquad Z'_{\mathcal{M}} \neq \bar{z}_{\mathcal{M}} \iff M_{\mathcal{M}} = 0 \text{ and } Z'_{\mathcal{M}} = Z_{\mathcal{M}},$$

where $0$ and $1$ are vectors of zeros and ones of appropriate shape.

Because $M$ is independent of $(Z, Y)$:

$$p(Y \mid Z'_{\mathcal{M}} = \bar{z}_{\mathcal{M}}, Z'_{-\mathcal{M}} = z_{-\mathcal{M}}) = p(Y \mid M_{\mathcal{M}} = 1, M_{-\mathcal{M}} = 0, Z_{-\mathcal{M}} = z_{-\mathcal{M}}) \qquad (2)$$
$$= p(Y \mid Z_{-\mathcal{M}} = z_{-\mathcal{M}}). \qquad (3)$$

Thus, knockout augmentation exactly corresponds to marginalization of the missing tool outputs.

---

1. As mentioned in the main paper, this is easily implemented by replacing the tool output $z_i$ with $\bar{z}_i$ of the same shape.

To see that $f_\theta$ learns all such marginals simultaneously, consider the expected training loss under tool knockout:

$$\mathcal{L}(\theta) = \mathbb{E}_{\boldsymbol{Z}',Y} \, \mathbb{E}_{\boldsymbol{M}} \, \ell\big(Y, \, f_\theta(\boldsymbol{Z}'(\boldsymbol{M}, \boldsymbol{Z}))\big) \tag{4}$$

$$= \mathbb{E}_{\boldsymbol{Z},Y} \, \mathbb{E}_{\boldsymbol{M}} \sum_{\boldsymbol{m} \in \boldsymbol{M}} \mathbb{I}(\boldsymbol{M} = \boldsymbol{m}) \, \ell\big(Y, \, f_\theta(\boldsymbol{Z}'(\boldsymbol{m}, \boldsymbol{Z}))\big) \tag{5}$$

$$= \mathbb{E}_{\boldsymbol{Z},Y} \sum_{\boldsymbol{m} \in \boldsymbol{M}} p(\boldsymbol{M} = \boldsymbol{m}) \, \ell\big(Y, \, f_\theta(\boldsymbol{Z}'(\boldsymbol{m}, \boldsymbol{Z}))\big) \tag{6}$$

$$= \sum_{\boldsymbol{m} \in \boldsymbol{M}} p(\boldsymbol{M} = \boldsymbol{m}) \, \mathbb{E}_{\boldsymbol{Z},Y} \, \ell\big(Y, \, f_\theta(\boldsymbol{Z}'(\boldsymbol{m}, \boldsymbol{Z}))\big), \tag{7}$$

where $\mathbb{I}$ is the indicator function. That is, the objective is a weighted sum of losses, each corresponding to estimating a conditional distribution in the support of $p(\boldsymbol{M})$.

## Appendix F. Prompts

We use MedGemma as a vision–language tool selector. For each image, modality, and task, we provide the model with (1) the toolbox $\mathcal{T} = \{t_1, \dots, t_N\}$ (2) a brief description of the modality and each tool (3) a task-specific natural language instruction

The model is asked to return a JSON object specifying which tools from $\mathcal{T}$ should be used for that image and task.

For Camelyon17 (histopathology), the toolbox consists of nuclei-level tools that identify nuclei centroids, bounding boxes, contours, types, and type probabilities: `histo_nuc_centroid`, `histo_nuc_bbox`, `histo_nuc_contour`, `histo_nuc_type`, and `histo_nuc_type_prob`

For ISIC 2017 (dermatology), the toolbox is constructed from a lesion segmentation tool, dermoscopic structure maps, and color-based markers: `derm_lesion_segmenter`, `derm_pigment_network`, `derm_negative_network`, `derm_milia_like_cyst_detector`, `derm_streaks_detector`, `derm_marker_malignant_union`, and `derm_marker_browns`

For provide the VLM with a toolbox consisting with the union of all tools across modalities.

```
TOOLBOX = {
    "histo_nuc_centroid",
    "histo_nuc_bbox",
    "histo_nuc_contour",
    "histo_nuc_type",
    "histo_nuc_type_prob",
    "derm_lesion_segmenter",
    "derm_pigment_network",
    "derm_negative_network",
    "derm_milia_like_cyst_detector",
    "derm_streaks_detector",
    "derm_marker_malignant_union",
    "derm_marker_browns"
}
```

For both fixed and dynamic tool selection, we also provide MedGemma with a brief natural language description of each tool:

```
TOOL_DESCRIPTIONS = {
  "histo_nuc_centroid":   "Returns each nucleus centroid in
                           [x_center, y_center]",
  "histo_nuc_bbox":       "Returns each nucleus bounding box in
                           [x_top_left, y_top_left, width, height]",
  "histo_nuc_contour":    "Returns the polygon points
                           tracing each nucleus boundary",
  "histo_nuc_type":       "Returns the predicted nucleus type label
                           (0-5; e.g., epithelial, inflammatory,
                           connective, dead, non-neoplastic
                           epithelial)",
  "histo_nuc_type_prob": "Returns class-probability scores for the
                            predicted nucleus type",

  "derm_lesion_segmenter":            "Segments lesion ROI",
  "derm_pigment_network":             "Detects reticular pigment
                                        network",
  "derm_negative_network":            "Detects negative network (white
                                        lines)",
  "derm_streaks_detector":            "Detects radial streaks or
                                        pseudopods at edges",
  "derm_milia_like_cyst_detector": "Detects milia-like cysts (often
                                      SK)",
  "derm_marker_malignant_union":    "Union of malignancy chromatic
                                      markers",
  "derm_marker_browns":               "Detects brown pigment regions"
}
```

The base prompt template for MedGemma in the fixed tool selection setting is:

```
You are a medical expert in {modality}. Select tools for a single
task from a fixed toolbox {TOOLBOX} described by {TOOL_DESCRIPTIONS
    }.
Choices must depend on the task and image evidence.

Choose max {max_tools} tools from the toolbox {TOOLBOX}
that are most relevant for solving the task in each image; no
duplicates.

Return ONLY JSON with the following fields:
- task_modality
- task
- selected_tools
- abstain  (boolean)

Each entry in selected_tools must include:
- id            (tool name from TOOLBOX)
- rank          (1..N, 1 = most important)
- confidence    (float in [0, 1])
```

```
- reason        (brief phrase tied to image cues)

If you are unsure about the modality or task, set task_modality =
"unknown" and abstain = true.

Return ONLY JSON.
```

We then append a task-specific instruction depending on the modality and classification problem. For the Camelyon17 and ISIC 2017 BM/MN tasks, we use the following prompt instructions:

```
Task: Determine if the central 32x32 region of this 96x96
histopathology patch contains tumor or not.
Answer ONLY in the following format:
label: tumor or no tumor
prob: <a real-valued prediction probability in [0,1] that the
predicted label is correct>
```

```
Task: Determine if the lesion in the dermoscopic image
is malignant or benign.
Answer ONLY in the following format:
label: malignant or benign
prob: <a real-valued prediction probability in [0,1] that the
predicted label is correct>
```

```
Task: Determine if the lesion in the dermoscopic image
is melanocytic or non-melanocytic.
Answer ONLY in the following format:
label: melanocytic or non-melanocytic
prob: <a real-valued prediction probability in [0,1] that the
predicted label is correct>
```

**Dynamic tool selection.**   For dynamic tool selection , we instead prompt MedGemma to score every tool within the given modality in the toolbox. Details of the dynamic selection procedure are described in Section D.3. The scoring prompt replaces the selection instructions above with

```
You are a medical expert in {modality}. Score each tool in the
provided toolbox {TOOLBOX} described by {TOOL_DESCRIPTIONS} between
[0,1]. Base scores strictly on the visible image evidence and task
   relevance.
Return JSON ONLY with keys: task_modality, task, scores.
- scores must be a list of objects with: id (string), score (integer
0...1).
- Provide exactly one score for each tool id you are given.
Scores do not need to be rounded numbers. No omission of scores.
```

The same toolbox, tool descriptions, and task-specific instructions are used as in the fixed-selection setting. The full prompt templates above correspond to the references in Section 3.2 and Appendix D.3.

### F.1. LLaVA-Med Prompts

For our LLaVA-Med zero-shot baseline, we use the following prompt instructions for Camelyon17, ISIC-BM, and ISIC-MN tasks:

```
Instruction:
You are a pathology assistant. Look at the central 32x32 region of
this 96x96 histopathology patch and answer strictly:
Is there tumor present in the central region or not?
Answer ONLY in the following format:
label: tumor or no_tumor
prob: <a real-valued prediction probability in [0,1] that the
predicted label is correct>
```

```
Instruction:
You are a dermatology expert. Look at this dermoscopic image of a
skin lesion.
Is the lesion benign or malignant?
Answer ONLY in the following format:
label: malignant or benign
prob: <a real-valued prediction probability in [0,1] that the
predicted label is correct>
```

```
Instruction:
You are a dermatology expert. Look at this dermoscopic image of a
skin lesion.
Is the lesion melanocytic or not?
Answer ONLY in the following format:
label: melanocytic or non-melanocytic
prob: <a real-valued prediction probability in [0,1] that the
predicted label is correct>
```

## Appendix G. Computational Cost

Table 4: We report parameter count, FLOPs, and inference time per image for a single forward pass. FLOPs are reported per image.

| Model | Params | FLOPs / image | Inference time (s) |
|---|---|---|---|
| TBM | 0.139 M | $3.37 \times 10^8$ | $5.22 \times 10^{-4}$ |
| TBF (VLM + Tools + TBM) | 4.34 B | $1.40 \times 10^{12}$ | 3.272 |
| EfficientNet-B0 | 4.011 M | $4.00 \times 10^8$ | $2.16 \times 10^{-3}$ |
| MedGemma (Zero-shot) | 4.34 B | $1.33 \times 10^{12}$ | 2.191 |

