# OpenReview forum: "A Tool Bottleneck Framework for Clinically-Informed and Interpretable Medical Image Understanding"
_MIDL.io/2026/Conference — MIDL 2026 Poster_

### Official Review · Reviewer_pK5i · 2026-01-07

**Confidence:** 3
**Preliminary Rating:** 4
**Final Rating:** 4

**Summary:**

This paper introduces an approach to medical image understanding that tackles the shortcomings of text-based tool composition found in current vision-language models. The authors work employs a medical VLM to select clinically relevant tools from a predefined toolbox, merging their outputs through a learned model. Experiments performed on histopathology and dermatology datasets reveal that their framework achieves comparable or even superior performance to black-box classifiers and leading tool-use frameworks. The importance of this framework lies in its capacity to deliver interpretable, clinically relevant predictions while also demonstrating significant performance improvements, particularly in scenarios with limited data. Their work builds sufficiently from related work and the flow of content is good right from the beginning of the paper. I got a good sense of what the authors are trying to tackle through their contributions in the paper.

**Strengths:**

The paper is well-structured, and includes good comparisons against a wide array of baselines. Their framework shows superior performance in low-data settings, making it highly valuable for medical domains where labeled data is scarce. The proposed random perturbation strategy during training improves the model's robustness. Figure 1 is a good addition to the paper and helps a lay reader understand their architecture better. The related work section is pretty comprehensive and helps one understand the background of the paper well.

**Weaknesses:**

1. Why is the framework second best on ISIC-BM from Table 1?
2. Are there any other datasets or metrics that you could have considered for evaluation in the experimental section?
3. I feel that the problem statement could be better formulated in the introduction section right before the part where you talk about the papers contributions.
4. Could you provide more details on how you would scale your work to use a more comprehensive set of tools as you mentioned in the future direction section?

**Detailed Comments:**

The detailed comments are as in the weaknesses section above.

**Justification Of Final Rating:**

Having read the authors thoughtful responses to my questions, I would like to maintain my score as a accept. As I saw it in the initial phase and upon reading the other reviewers comments I didn't see any strong weaknesses to lower my score. The authors did great on their detailed point by point responses.

**Justification Of The Preliminary Rating:**

I didn't see any major weaknesses here and I felt that this work is both timely and technically robust, addressing gaps found in current medical VLMs. The language and writing was also pretty fluent helping one get a sense of the contributions of the paper.

**Questions To Address In The Rebuttal:**

The questions are in the weaknesses section above.

---

> ### Author Response · Authors · 2026-01-25
> **Response to Reviewer pK5i**
>
> We thank Reviewer pK5i for their helpful comments and suggestions for improvement. We provide a point-by-point response below.
>
> > Why is the framework second best on ISIC-BM from Table 1?
>
> While we concede that TBF is not the best-performing in terms of quantitative metrics across all tasks we considered, TBF has several advantages beyond the best-performing model which are especially important in medical imaging contexts.
>
> TBF exhibits superior clinical grounding than EfficientNet. Since EfficientNet is a black-box model that is trained end-to-end on raw images, it does not encode inductive biases that are catered towards the task at hand. In contrast, since TBF bottlenecks the prediction through tools that extract task-relevant clinical features, this encodes task-specific inductive biases. In particular, we find that this leads to improvement in data-limited scenarios, as we demonstrate in Section 5.2.
> TBF exhibits superior interpretability as compared to EfficientNet. This is related to our point above about clinical grounding. In addition, we show in Section 5.3 how we can quantify a notion of “tool importance”, which gives an explicit weight to each tool for interrogating the model’s decision-making. Note that no other baselines have the ability to output this type of tool importance.
>
>
>
>
> > I feel that the problem statement could be better formulated in the introduction section right before the part where you talk about the papers contributions.
>
> Thank you for your suggestion, we have made this change in the revised paper. In particular, we have added the following sentence to the paragraph before the paper contributions:
>
> “In particular, ours is the first work to incorporate clinically-grounded priors for tool use using a neural-network based composition”
>
> > Could you provide more details on how you would scale your work to use a more comprehensive set of tools as you mentioned in the future direction section?
> > Are there any other datasets or metrics that you could have considered for evaluation in the experimental section?
>
> Thank you for these comments. We believe that TBF has wide applicability to many different tasks, modalities, and data types in medical imaging contexts. In a clinical setting, we envision a comprehensive toolbox of hundreds or thousands of tools each capturing different relevant clinical features across different modalities. These tools would target different modalities, anatomies of interest, and subpopulations. Note that in this scenario, our core setup remains unchanged – the VLM still select from this vast list of tools the most relevant and the TBM is trained to be robust to such tool selections during training.
>
> Additionally, in response to additional datasets and metrics, we added an additional experiment where we performed 3-way classification on the ISIC dataset. The task is to classify between Melanoma, Nevus, and Seborrheic keratosis image-level labels. We follow the same experimental setup as our other ISIC experiments in Section 5.2, as we are particularly interested in the data-limited regime. Due to the time-constraint, we were only able to experiment with the two best-performing baselines, EfficientNet and Y-Net (the tool-use baselines require significant computational overhead). We train both TBF and EfficientNet on 64 images each, and evaluate on the full test set. We report the one-versus-rest AUC results below:
>
> | EfficientNet | Y-Net | TBF |
> |----------------------------------------|---------------------|----------------------------|
> | 0.68               | 0.65| 0.72                         |

---

### Official Review · Reviewer_bKko · 2026-01-10

**Confidence:** 4
**Preliminary Rating:** 2
**Final Rating:** 4

**Summary:**

The paper introduces the Tool Bottleneck Framework(TBF), which uses a vision–language model to select clinically relevant tools and a learned Tool Bottleneck Model to fuse their spatial outputs for medical image prediction. By replacing text-based tool composition with neural feature fusion, the approach improves interpretability and robustness. Experiments on histopathology and dermatology benchmarks show that TBF matches or outperforms strong CNN and tool-use baselines, especially in low-data settings.

**Strengths:**

1. The proposed Tool Bottleneck Framework explicitly integrates clinically meaningful tools and enables tool-level importance analysis, offering clearer interpretability than end-to-end CNNs or text-based tool-use methods.
2. The method demonstrates consistent performance gains or parity across histopathology and dermatology benchmarks, with particularly notable improvements when training data is scarce, highlighting its data efficiency.

**Weaknesses:**

1. Although the author emphasizes clinically informed “reasoning,” the proposed approach primarily performs learned feature fusion over tool outputs. The work lacks clear analysis or ablations to demonstrate reasoning behavior beyond structured feature aggregation.
2. For the ISIC experiments, ground-truth dermoscopic structure annotations are directly used as tool outputs, effectively providing near-oracle information. This setting likely overestimates performance and raises concerns about applicability in realistic clinical deployments.
3. Comparisons with CNNs and Y-Net may not fully control for differences in supervision and inductive bias, as the proposed method benefits from explicit pixel-level clinical priors encoded in tools, potentially confounding the source of performance gains.

**Detailed Comments:**

Minor issues:
1. The paper defers most details of the VLM prompting strategy and tool-selection format to the appendix. Providing a concise example prompt and selection output in the main text would improve clarity and reproducibility.

**Justification Of Final Rating:**

The rebuttal has addressed most of my concerns. After revision, the work is clear and technically sound to me. Acknowledging the increased quality of the revised work, I will raise my score accordingly.

**Justification Of The Preliminary Rating:**

The work overstates its claims of “reasoning,” relies on idealized tool assumptions in parts of the evaluation, and lacks sufficient analysis of robustness and realism in tool selection and deployment. While the approach is promising and technically sound, these limitations reduce confidence in its practical impact and conceptual novelty, leading to a weak reject recommendation.

**Questions To Address In The Rebuttal:**

Please refer to the weakness and detailed comments.

---

> ### Author Response · Authors · 2026-01-24
> **Response to Reviewer bKko**
>
> We thank Reviewer bKko for their comments and suggestions for improvement. We provide a point-by-point response below.
>
> > For the ISIC experiments, ground-truth dermoscopic structure annotations are directly used as tool outputs, effectively providing near-oracle information. This setting likely overestimates performance and raises concerns about applicability in realistic clinical deployments.
>
> Thank you for this comment. Prompted by this suggestion, we have replaced our previous use of ground-truth dermoscopic annotations with an open-source model from Kawahara et al. [1] to serve as tools for predicting all four dermoscopic features. This effectively removes our dependence on ground truth annotations and more closely aligns with our model’s intended use case.
>
> We report our results below. Notably, we find that ISIC-BM improves by 0.5% percent, while MN remains relatively unchanged. Thus, we conclude that our original takeaways remain unchanged.
>
> | Annotations as Tool Outputs (original) | Open-source model [1] as Tool Outputs (revised) |
> |----------------------------------------|-------------------------------------------------|
> | 77.0 (BM), 91.8 (MN)                   | 77.5 (BM), 91.7 (MN)                            |
>
>
> Details on this dermoscopic feature tool are included in Appendix C.2. ISIC experiments involving these dermoscopic features have been updated in the revised manuscript, and the results in Table 1 have been updated.
>
> > Comparisons with CNNs and Y-Net may not fully control for differences in supervision and inductive bias, as the proposed method benefits from explicit pixel-level clinical priors encoded in tools, potentially confounding the source of performance gains.
>
> Prompted by this, we have added two additional baselines: Gemma Zero-Shot and Gemma w/ Tool Prompts (with pixel level tool outputs) to Table 1. We include the revised table below and in the revised manuscript:
>
> | Model                               | Camelyon17 | ISIC-BM | ISIC-MN |
> |-------------------------------------|------------|---------|---------|
> | Gemma (Team et al., 2025)           | 50.0       | 49.5    | 50.4    |
> | Gemma w/ Tool Prompts               | 50.9       | 47.1    | 48.8    |
> | MedGemma (Sellergren et al., 2025)  | 50.0       | 44.4    | 48.7    |
> | MedGemma w/ Tool Prompts            | 50.0       | 46.8    | 50.0    |
> | VisProg (Gupta and Kembhavi, 2023)  | 50.4       | 50.0    | 50.0    |
> | LlavaMed (Li et al., 2023)          | 50.0       | 49.4    | 50.0    |
> |                                     |            |         |         |
> | EfficientNet (Tan and Le, 2019)     | 88.6       | **78.4** | 91.2 |
> | Y-Net (Mehta et al., 2018)          | 88.2       | 65.8    | 86.6    |
> | LlavaMed FT                         | 66.2       | 51.5    | 58.0    |
> |                                     |            |         |         |
> | **TBF (ours)**                      | **92.3**   | 77.5    | **91.7** |
> | → without perturbation (α = 1)      | −2.1       | −2.1    | −1.4    |
> | → with all modality-specific tools  | −0.2       | −2.2    | −0.7    |
>
> In particular, we find that Gemma Zero-Shot and Gemma w/ Tool Prompts perform similarly to other VLMs and tool-use frameworks. As we discussed in the original manuscript, we identify the main shortcoming of these methods as composing tool outputs via text, which fundamentally cannot compose the fine-grained detail within the tool outputs. TBF is, to the best of our knowledge, the only model that uses a neural network to do tool composition – we see that our conclusions of the superiority of TBF in our experiments remains unchanged.
>
> To clarify, our Y-Net baseline and all tool-use baselines (MedGemma w/ Tools and VisProg) also incorporate the same amount of supervision and pixel-level clinical priors via tools. We believe that the ability for TBF to outperform these baselines despite the same amount of supervision is due to our proposed approach of leveraging neural network-based composition rather than text-based composition. In particular, we would like to clarify that our proposed approach of incorporating pixel-level clinical priors encoded via tools is a contribution of our work. To the best of our knowledge, no prior work has proposed a neural network-based composition of tools as a way to incorporate clinical priors. To make this more clear, we have added the following sentence to our contributions portion of the introduction in the revised manuscript:
> “In particular, ours is the first work to incorporate clinically-grounded priors for tool use using a neural-network based composition.”
>
> [1] Jeremy Kawahara and Ghassan Hamarneh. Fully convolutional neural networks to detect
> clinical dermoscopic features. IEEE Journal of Biomedical and Health Informatics, 23
> (2):578–585, 2019. ISSN 21682194. doi: 10.1109/JBHI.2018.2831680.

---

> > ### Author Response · Authors · 2026-01-24
> > **Response to Reviewer bKko (continued)**
> >
> > > The paper defers most details of the VLM prompting strategy and tool-selection format to the appendix. Providing a concise example prompt and selection output in the main text would improve clarity and reproducibility
> >
> > Thank you for this suggestion. We have added an example prompt and selection output in Section 3.2 “Vision-Language Model as a Tool Selector” for improving clarity and reproducibility in the main text. In particular, we include the following prompt in Section 3.2:
> >
> > “An example prompt is provided below, and an example VLM-selection output is provided in Figure 1.”
> >
> > > Although the author emphasizes clinically informed “reasoning,” the proposed approach primarily performs learned feature fusion over tool outputs. The work lacks clear analysis or ablations to demonstrate reasoning behavior beyond structured feature aggregation.
> >
> > Thank you for this important comment. In the context of our work, baselines like VisProg are referred to as performing “visual reasoning” in the sense that they decompose an image understanding task into multiple subtasks (“tools”) and then compose these subtasks to make a prediction. These baselines perform this decomposition by using a VLM to output text that embeds function calls for the tools. Similarly, text is used to subsequently compose these tool outputs. We argue that TBF works in a similar way, except that it uses a learned neural network to compose the tool outputs. This overcomes the limitation of text-based composition, which fundamentally cannot compose fine-grained details present in tool outputs for medical images.
> >
> > In addition, we argue that our formulation enables one to interrogate model reasoning via tool importance (Section 5.3). In particular, in Figure 3 left, we observe that for predicting cancer vs. benign tissue, our model “reasons” about its decision by placing a high importance (about 0.04) on nucleus contours and low importance (near 0) on nucleus type probability. Note that no other baselines have the ability to output this type of reasoning.
> >
> > To avoid any potential for confusion in the future, we have replaced the phrase “visual reasoning” with “visual understanding” in the revised manuscript.

---

### Official Review · Reviewer_F7oN · 2026-01-16

**Confidence:** 4
**Preliminary Rating:** 4
**Final Rating:** 4

**Summary:**

The paper introduces the Tool Bottleneck Framework (TBF), an approach designed to enhance interpretability and performance in medical image understanding. TBF employs a Vision-Language Model (VLM) to select a subset of clinically relevant tools for a given image and task. Experiments on histopathology (Camelyon17) and dermatology (ISIC) datasets demonstrate that TBF achieves performance on par with or superior to state-of-the-art VLMs and deep learning baselines.

**Strengths:**

The proposed Tool Bottleneck Model (TBM) demonstrates strong utility for the medical domain by outperforming baselines in data-limited regimes. The framework inherently supports interpretability through its "tool" strategy. paper is well-written and logically structured, effectively contrasting the proposed method with "black-box" approaches to justify the architectural choices.

**Weaknesses:**

1. The ablation study in Table 1 reveals that the "all modality-specific tools" baseline performs very similarly to the proposed TBF with VLM selection (e.g., 92.1% vs 92.3% on Camelyon17). This suggests that the VLM's role in selecting tools adds minimal performance gain over simply fusing all available tools. The complexity of integrating a VLM might not be justified if a static ensemble of tools works equally well.
2. The evaluation is restricted to binary classification tasks (Tumor vs. Normal; Benign vs. Malignant). Medical imaging often involves multi-class classification or segmentation tasks. It is unclear if the TBF framework scales effectively to more complex output spaces.

**Detailed Comments:**

While the paper mentions data efficiency during training, it does not discuss inference cost. Running a VLM + multiple image processing tools + the TBM seems computationally expensive compared to a single EfficientNet. A discussion on computational cost would add value.

**Justification Of Final Rating:**

I would like to thank the authors for their response and the effort put into running additional experiments during the rebuttal period. The authors have effectively addressed my major concerns. I maintain my rating of 4.

**Justification Of The Preliminary Rating:**

The paper proposes a novel framework (TBF), and the results on data efficiency are compelling for the medical domain. However, the marginal performance gain of the VLM selection over using a static set of all tools raises questions about the necessity of the complex selection component.

**Questions To Address In The Rebuttal:**

1. Please explain more about the results of ablation study in Table 1.
2. Is TBM suitable for multi-class classification? Current experiments are on binary classification tasks.
3. What is the inference cost for TBM? Especially compared with other baseline models.
4. Please add more baselines including general LLMs e.g. GPT/Gemini and medical-expert models e.g. Baichuan.

---

> ### Author Response · Authors · 2026-01-25
> **Response to Reviewer F7oN**
>
> We thank Review F7oN for their helpful comments and suggestions for improvement. We provide a point-by-point response below.
>
> > While the paper mentions data efficiency during training, it does not discuss inference cost. Running a VLM + multiple image processing tools + the TBM seems computationally expensive compared to a single EfficientNet. A discussion on computational cost would add value.
> What is the inference cost for TBM? Especially compared with other baseline models.
>
> Thank you for this important comment. Prompted by this, we have added an additional section, Appendix G. “Computational Cost”, which provides a Table that details the parameter count, inference time in seconds, and FLOPs for each baseline.
>
> We include this table below:
> | Model                         | Params   | FLOPs / image     | Inference time (s) |
> |------------------------------|----------|--------------------|---------------------|
> | TBM                          | 0.139 M  | 3.37 × 10^8        | 5.22 × 10^-4        |
> | TBF (VLM + Tools + TBM)      | 4.34 B   | 1.40 × 10^12       | 3.272               |
> | EfficientNet-B0              | 4.011 M  | 4.00 × 10^8        | 2.16 × 10^-3        |
> | MedGemma (Zero-shot)         | 4.34 B   | 1.33 × 10^12       | 2.191               |
>
>
>
>
> > Please add more baselines including general LLMs e.g. GPT/Gemini and medical-expert models e.g. Baichuan.
>
> We have added two additional baselines to our revised manuscript: Gemma Zero-Shot and Gemma w/ Tool Prompts (with pixel level tool outputs) to Table 1. We include the revised table below and in the revised manuscript:
>
> | Model                               | Camelyon17 | ISIC-BM | ISIC-MN |
> |-------------------------------------|------------|---------|---------|
> | Gemma (Team et al., 2025)           | 50.0       | 49.5    | 50.4    |
> | Gemma w/ Tool Prompts               | 50.9       | 47.1    | 48.8    |
> | MedGemma (Sellergren et al., 2025)  | 50.0       | 44.4    | 48.7    |
> | MedGemma w/ Tool Prompts            | 50.0       | 46.8    | 50.0    |
> | VisProg (Gupta and Kembhavi, 2023)  | 50.4       | 50.0    | 50.0    |
> | LlavaMed (Li et al., 2023)          | 50.0       | 49.4    | 50.0    |
> |                                     |            |         |         |
> | EfficientNet (Tan and Le, 2019)     | 88.6       | **78.4** | **91.2** |
> | Y-Net (Mehta et al., 2018)          | 88.2       | 65.8    | 86.6    |
> | LlavaMed FT                         | 66.2       | 51.5    | 58.0    |
> |                                     |            |         |         |
> | **TBF (ours)**                      | **92.3**   | 77.5    | **91.7** |
> | → without perturbation (α = 1)      | −2.1       | −2.1    | −1.4    |
> | → with all modality-specific tools  | −0.2       | −2.2    | −0.7    |
>
> In particular, we find that Gemma Zero-Shot and Gemma w/ Tool Prompts perform similarly to other VLMs and tool-use frameworks. As we discussed in the original manuscript, we identify the main shortcoming of these methods as composing tool outputs via text, which cannot compose the fine-grained detail within the tool outputs for medical images. TBF is, to the best of our knowledge, the only model that uses a neural network to do tool composition – we see that our conclusions of the superiority of TBF in our experiments remains unchanged.
>
> > The ablation study in Table 1 reveals that the "all modality-specific tools" baseline performs very similarly to the proposed TBF with VLM selection (e.g., 92.1% vs 92.3% on Camelyon17). This suggests that the VLM's role in selecting tools adds minimal performance gain over simply fusing all available tools. The complexity of integrating a VLM might not be justified if a static ensemble of tools works equally well.
>
> We would like to clarify that our vision is that in a clinical setting, the toolbox will contain potentially hundreds or thousands of tools related to a variety of clinical features. These tools would target different modalities, anatomies of interest, and subpopulations. In this case, fusing all available tools is computationally intractable, and thus we view that ablation as an academic exercise. To clarify this, we have added the following to the revised manuscript:
>
> “Second, we ablate the VLM tool selector and simply pass in all modality-specific tools. Note that in the case of large $N$, this is computationally intractable.”

---

> > ### Author Response · Authors · 2026-01-25
> > **Response to Reviewer F7oN (continued)**
> >
> > > Please explain more about the results of ablation study in Table 1.
> >
> > Thank you for this clarifying point. The first row of the ablations in Table 1 ablates the perturbation of VLM tool selections described in Section 3.3; this corresponds to using a value of α = 1. That is, we do not inject additional randomness to the tool selections, thus limiting the number of tool combinations seen by the TBM during training. We observe that this perturbation is important for increasing robustness of the TBM to different subsets of tools.
> >
> > The second row of the ablations in Table 1 ablates the VLM tool selector entirely, and simply passes in all modality-specific tools. We would like to clarify that our vision is that in a clinical setting, the toolbox will contain potentially hundreds or thousands of tools related to a variety of clinical features. In this case, fusing all available tools is computationally intractable, and thus we view that ablation as an academic exercise.
> >
> > Prompted by this comment, we have added additional details in Section 5.1 describing the ablations. In particular, we have added these two sentences:
> >
> > “First, we ablate the perturbation of VLM tool selections described in Section 3.3; this corresponds to using a value of α = 1.”
> >
> > “Second, we ablate the VLM tool selector and simply pass in all modality-specific tools.”
> >
> >
> > > The evaluation is restricted to binary classification tasks (Tumor vs. Normal; Benign vs. Malignant). Medical imaging often involves multi-class classification or segmentation tasks. It is unclear if the TBF framework scales effectively to more complex output spaces
> > Is TBM suitable for multi-class classification? Current experiments are on binary classification tasks.
> >
> > Prompted by this comment, we have added an additional experiment where we performed 3-way classification on the ISIC dataset. The task is to classify between Melanoma, Nevus, and Seborrheic keratosis image-level labels. We follow the same experimental setup as our other ISIC experiments in Section 5.2, as we are particularly interested in the data-limited regime. Due to the time-constraint, we were only able to experiment with the best-performing baselines, EfficientNet and Y-Net. We train both TBF and EfficientNet on 64 images each, and evaluate on the full test set. We report the one-versus-rest AUC results below:
> >
> > | EfficientNet | Y-Net | TBF |
> > |----------------------------------------|--------------------------|-----------------------|
> > | 0.68               | 0.65| 0.72                         |
> >
> > Note that the tool-use baselines require significant computational overhead and we were not able to include them as results by the rebuttal deadline. However, we do not foresee them outperforming any of the above three models, given our other binary classification experiments on ISIC. Once completed, we plan to add these comprehensive results to the revised manuscript.

---

> > > ### Comment · Reviewer_F7oN · 2026-02-02
> > >
> > > I would like to thank the authors for their response and the effort put into running additional experiments during the rebuttal period. The authors have effectively addressed my major concerns. I maintain my rating of 4.

---

### Author Rebuttal · Authors · 2026-01-25

**Rebuttal:**

We attach our revised manuscript with changes in red as Supporting Material.

**Supporting Material:**

/attachment/09af4319ba03662c17da4f9980e38301b1731692.pdf

---

### Comment · Area_Chair_aBT7 · 2026-02-01
**Reminder: Please Update Final Rating (Feb 1, AoE)**

Dear Reviewers,

Thank you for your reviews and participation in the discussion. As we are approaching the end of the discussion period, I kindly ask you to please update your Final Rating by going to Edit → Official Review and entering your final score by February 1st, 2026 (23:59 AoE).

If applicable, please also briefly respond to the authors’ rebuttal and engage with other reviewers’ comments, especially where there are differing opinions (which we have in the case of this paper). This will be very helpful for the meta-review and final decision process.

Thank you again for your time and contribution.

---

### Meta-Review · Area_Chair_aBT7 · 2026-02-08

**Recommendation:** Accept (Poster)
**Confidence:** 5

**Metareview:**

The reviewers reached a consensus that the proposed framework offers a novel and technically sound solution for incorporating clinical tools into VLM pipelines, particularly in low-data regimes. The authors successfully addressed concerns regarding data leakage and baseline comparisons during the rebuttal, confirming the method's robustness and utility for the medical imaging community.

---

### Decision · Program_Chairs · 2026-02-14

Accept (Poster)